# How urban environment shapes EV charging experience in Travis County, Texas

**Ahyoung Chang** **\*, Seung Gyu Baik, Junfeng Jiao\***

Urban Information Lab, The University of Texas at Austin, Austin, Texas, United States of America

\* ahyoung@utexas.edu (AC); jjiao@austin.utexas.edu (JJ)

## Abstract

As electric vehicle charging stations (EVCSs) become increasingly embedded in cities, understanding user experience is critical for designing infrastructure that is functional, accessible, and comfortable. This study integrates AI-driven sentiment analysis with spatial modeling to examine how urban environments shape EVCS perception. Using more than 4,000 user reviews from Travis County, Texas, three large language models classified sentiment across categories such as charging operation, accessibility, and parking. Random Forest regression results show that walkability, greenery, openness, and surrounding amenities are among the strongest predictors of user sentiment, while operational concerns are widely distributed in peripheral areas and accessibility and parking frustrations cluster in dense commercial zones. These findings demonstrate that EVCS usability is influenced not only by technical performance but also by contextual qualities of the built environment. This work provides a scalable framework for real-time monitoring of EVCS experiences and actionable insights for location-sensitive planning that supports user wellbeing and sustainable mobility adoption.

## Introduction

As electric and autonomous vehicle adoption accelerates, Electric Vehicle Charging Stations (EVCSs) have become central to modern urban mobility systems [1]. Beyond their technical role in energy transfer, EVCSs shape user behavior, mobility patterns, and local economic activity. Their location and surrounding environment influence not only energy use but also pedestrian flows, commercial dynamics, and overall urban experience [2,3]. Because EVCSs are so deeply embedded in the urban fabric, evaluating their success requires looking beyond foundational engineering metrics to understand the human-centered experience.

While recent research has increasingly recognized this broader role, most studies have continued to prioritize macro-level factors such as accessibility, spatial distribution, and energy capacity [4–7]. Consequently, the spatially varied and subjective nature of user experience remains underexplored. Satisfaction depends not only

**Data availability statement:** All dataset and author-generated Python code files used in this study are publicly available from the OpenICPSR repository (URL: https://www.openicpsr.org/openicpsr/project/246962/version/V1/view).

**Funding:** This study was financially supported by the MITRE Corporation in the form of a Good Systems MITRE grant awarded to JJ. This study received additional financial support from the University of Texas at Austin Good Systems Grand Challenges Program in the form of a Good Systems Smart City Funding grant awarded to JJ. Further financial support for this study was provided by the National Science Foundation in the form of grants awarded to JJ (2125858, 2236305). The funders had no role in study design, data collection and analysis, decision to publish, or preparation of the manuscript.

**Competing interests:** The authors have declared that no competing interests exist.

on operational reliability, but also on contextual qualities such as walkability, openness, and proximity to amenities. Capturing this human-centered dimension requires tools that connect nuanced user feedback with measurable features of the built environment.

In prior work, we demonstrated how generative AI models could analyze over 4,000 user-generated reviews to evaluate EVCS sentiment patterns and reveal spatial variations in user experiences [8]. While this provided a methodological foundation for AI-driven sentiment and spatial analysis, it left a critical question open: which specific urban environmental factors most strongly shape user satisfaction?

Building on that foundation, this study advances a two-pronged framework that combines natural language and spatial data. First, we apply Generative AI (GenAI) models to classify sentiment and thematic content in EVCS reviews, and to extract and quantify environmental features from Google Street View (GSV) panorama images. Second, we integrate these results with Random Forest regression to examine how urban form, visual context, and demographic variables influence user sentiment across categories such as accessibility, parking, operations, and pricing.

By linking unstructured perceptions with structured spatial indicators, this research moves beyond sentiment classification to identify the environmental determinants of EVCS usability. The findings highlight how features such as sidewalks, greenery, and amenity density contribute to perceived accessibility and satisfaction, offering new insights for designing charging infrastructure that is both technically reliable and spatially responsive.

The remainder of this paper proceeds as follows: the Literature Review section reviews relevant literature; the Materials and Methods section describes the data and methods; the Results section presents the analysis; and the Discussion and Conclusion section concludes with design implications and future research directions.

## Literature review

### Advances in AI-based methods for urban and geoscience research

Recent advances in GenAI and Large Language Models (LLMs) are reshaping urban and geoscience research by offering greater flexibility and reasoning capabilities than traditional machine learning (ML) models. Unlike ML models designed for narrow tasks such as traffic prediction or satellite image classification, LLM-based agents can perform diverse tasks, including data analysis, simulation, and decision support, through natural language and multimodal inputs [9,10].

GenAI-based agents have also been shown to simulate lifelike behavior in virtual cities [11], while Ma et al. (2024) integrated AI agents into mobility systems to enhance smart city operations [12]. One promising application is sentiment analysis of user-generated content such as reviews or social media posts to detect user needs and infrastructure issues in real time. This approach offers a human-centered lens for evaluating urban amenities, including EV charging stations [13,14]. Jiao and Chang (2025) applied multiple AI agent models to analyze EV charging station reviews. Their study compared the relative strengths of different AI models, and demonstrated that LLMs can effectively capture spatially embedded sentiment patterns by revealing

how operational issues were more common in suburban areas while accessibility and parking concerns clustered in dense commercial districts [8].

Despite this progress, important questions remain. Prior research has shown that GenAI models can detect spatially embedded sentiment patterns, but it is still unclear whether these AI-based interpretations truly reflect the physical qualities of urban environments. Little is known about how user perceptions captured through GenAI analysis interact with measurable features of the urban environment. Furthermore, while recent sentiment analysis application [8,13] have successfully demonstrated model performance and spatial clustering of sentiment, they offer limited insight into the causal mechanisms linking user experience to urban context. Addressing these gaps requires an integrated framework that systematically relates it to the physical and social characteristics of urban space.

## EV charging infrastructure and service

EVCSs are pivotal in the shift toward sustainable mobility. Most existing research has centered on power distribution, accessibility, and socioeconomic impacts, with emerging interest in AI-driven applications for demand forecasting and station optimization.

A key research stream explores the relationship between charging demand and electricity distribution [4,5,15]. Meng et al. (2022) proposed energy management frameworks that coordinate EVCSs with renewable sources [4], while Li and Jenn (2021) addressed grid congestion risks from clustered EV charging [5].

Another major focus is driver behavior and station accessibility [6,7,16–18]. Studies show that socio-demographics and urban form influence charging patterns, and that peak usage times should guide EVCS availability. Strategic station placement and service design have also been studied. Ren et al. (2022) incorporated land use and traffic data to optimize locations [7], while Asensio et al. (2023) emphasized the role of accessibility and service quality in user satisfaction [6]. Beyond infrastructure, recent studies investigate economic impacts. EVCS' presence has been linked to increased property values [19] and higher revenues for local businesses and hotels [2,3].

In recent literature, AI-driven approaches have emerged as experimental tools for demand forecasting and system monitoring. Qu et al. (2024) introduced a semantic model, ChatEV, for integrated demand prediction [20]. Yi et al. (2023) and Zhang et al. (2023) developed deep learning models based on real-world usage data to inform deployment and grid planning [21,22]. Jiao and Chang (2025) showed that GenAI models can be applied to large-scale EVCS reviews to classify sentiment and identify spatial variations in user experiences [8].

While prior research has focused on technical or locational aspects or provided a foundation for understanding user-centered perspectives, it did not fully address how such perceptions are shaped by quantitative features of the built environment. This study extends the scope of EVCS research by examining which aspects of urban context such as walkability, openness, amenity density and other variables.

## Urban environments matter to human perception

User experience is increasingly recognized as a key factor in the adoption and effectiveness of EVCSs. While charging speed and infrastructure coverage remain important, studies show that subjective perceptions, such as ease of use, pricing transparency, and reliability, strongly influence user satisfaction and willingness to use EVCSs. Beyond hardware performance, user experience shapes long-term engagement with charging networks. Integrating user feedback into infrastructure design enables more sustainable and responsive service provision [9].

A growing body of research emphasizes that urban environments themselves are deeply entangled with human emotional and cognitive responses [23–25]. Sadeghi et al. (2022) empirically demonstrated that subjective well-being is significantly associated with how residents perceive their neighborhood environment. Their findings, drawn from surveys in a historical Iranian urban context, show strong positive correlations between perceived neighborhood quality and indicators

such as social inclusion, life satisfaction, and mental well-being [24]. This supports the notion that built environments are not just physical containers of infrastructure but deeply embedded in the affective experience of urban life.

Furthermore, Li et al. (2025) synthesized a range of interdisciplinary studies integrating physiological signals, subjective perception data, and machine learning techniques [23]. Their review highlights how human perception of built environments can be meaningfully captured, predicted, and simulated, offering insight into how environmental stimuli (e.g., form, lighting, openness) affect neural and emotional responses. These machine-driven approaches open up novel pathways to evaluate infrastructure not merely in terms of function, but also in terms of human-centered experiential quality.

These insights are especially relevant for EVCSs, where waiting time, safety, and surrounding amenities all shape user experience. Sentiment analysis of user-generated reviews reveals that perceptions of safety, walkability, and comfort in the charging environment often surface alongside operational feedback [6,8]. This study builds on these understandings by applying sentiment analysis to EVCS reviews, identifying the most salient factors behind user satisfaction or frustration. With GenAI, this study aims to classify user review sentiments of EVCS experience and urban environmental features, explore whether and how these two dimensions are interrelated, and identify which contextual factors exert stronger influence across different types of human experiences.

## Materials and methods

To provide a clear overview of our research design, Fig 1 illustrates the methodological framework of this study. The analytical pipeline consists of four main phases: data collection and input, GenAI-based feature extraction, spatial integration, and machine learning–based spatial analysis. The following subsections detail each of these procedures.

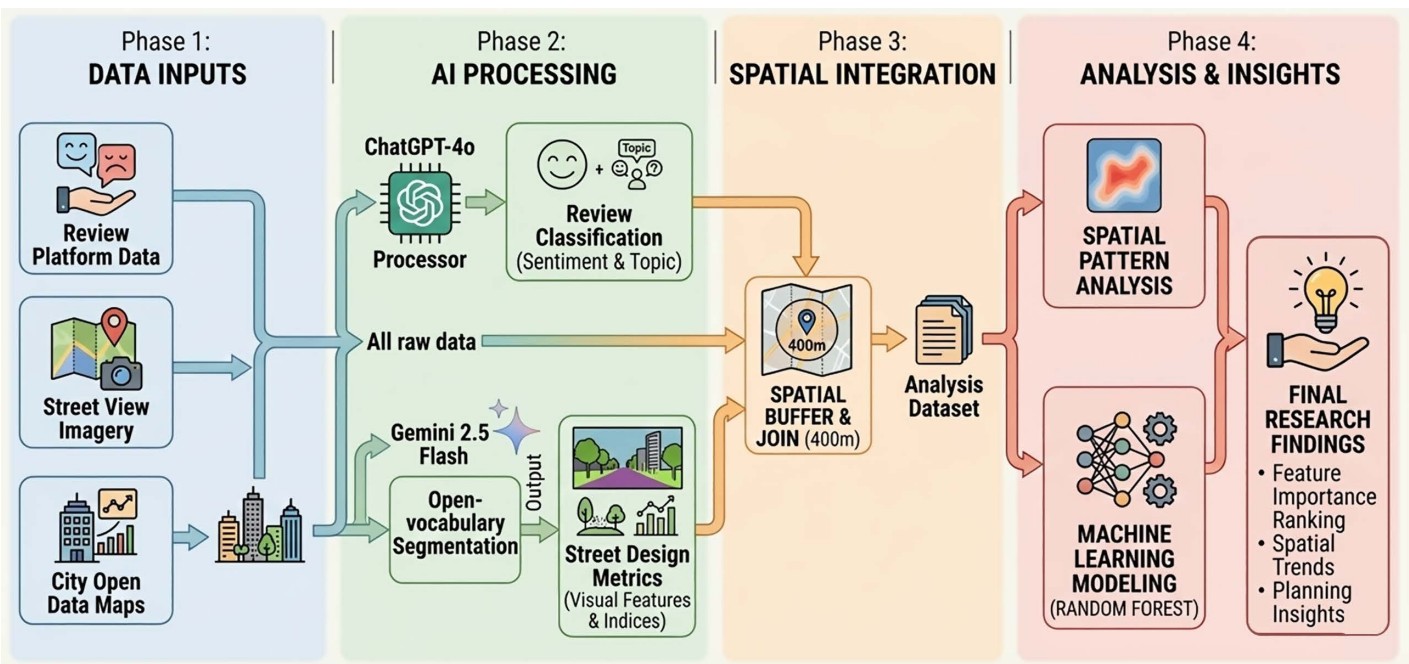

**Fig 1. Simplified methodological framework of the study.**

## Data collection and processing

**Study area.** Travis County covers roughly 1,023 square miles (2,650 km²) and has a population exceeding 1.3 million. As one of the nation's fastest-growing metropolitan areas, Austin has positioned itself at the forefront of technology, innovation, and sustainable mobility, making it a prime setting for examining EV charging infrastructure [26,27]. The city leads Texas in EV adoption, bolstered by Austin Energy's extensive public charging network and progressive clean transportation policies. Tesla's relocation to the region, combined with strong local climate initiatives, has further accelerated this growth [28]. While the broader Austin Metropolita Statistical Area encompasses multiple counties with highly fragmented policy landscapes and sparse infrastructure, we specifically selected Travas County to capture a distinct spatial gradient within a relatively cohesive administrative boundary. Travis County encompasses not only the dense, amenity-rich urban core of Austin, but also rapidly expanding suburban municipalities and sprawling, underserved rural peripheries in its eastern and western edges. This stark urban-to-rural heterogeneity provides a methodologically controlled yet diverse built environment. Consequently, Travis County provides a representative context for analyzing EVCS accessibility, user experience, and spatial variability [8]. These conditions make it a compelling case study for investigating the interplay between EV infrastructure, user sentiment, and broader urban planning strategies in the U.S.

**Data overview.** This study uses data from PlugShare, a widely used driver reporting platform that provides real-world insights into charging accessibility and reliability. PlugShare data include charger locations, specifications, nearby amenities, parking availability, PlugScore, and user reviews, which are essential for sentiment analysis. Prior research has validated PlugShare as a reliable source for estimating EVCS availability and user experiences at a granular level [8,29,30], making it particularly well-suited for this study.

To collect this information, we employed web scraping using Python libraries Selenium and BeautifulSoup, which systematically extracted details such as location, capacity, plug type, parking availability, nearby amenities, network brand, and user reviews for all EVCSs in Travis County [31–33]. The EVCSs included in this dataset are publicly accessible stations. Although the majority of these stations are owned and operated by private commercial network, such as ChargePoint, Tesla, and EVgo, they are open to the general public for use. Restricted residential or workplace-only chargers were excluded from the analysis. The final dataset, completed on January 8, 2025, contained 363 EVCS locations and 4,098 user reviews. Reviews are categorized by PlugShare into Charging Completed, Comments, and Errors, though only a subset is sentiment relevant.

This study collected data of visual imagery from Google Maps to systematically evaluate the surrounding environments of EVCSs. The imagery data were gathered using the Google Maps API to extract Street View Panorama images within 400m radius of each EVCS. Panoramas were retrieved from locations such as major intersections, public plazas, and key radiating streets to capture a representative and diverse view of the urban environment as experienced by EVCS users. To quantitatively evaluate the relationship between the user experience of EVCS and its surrounding urban environment, this study introduced the following proxies: land use, points of interest, economic self-sufficiency, and street design elements. These specific proxies were selected because prior urban planning and transportation literatures have consistently shown that macro-level functional destinations, such as land use and POIs, socio-spatial equity, and micro-level visual comfort (e.g., street design elements) are primary determinants of human behavior, safety perceptions, and overall satisfaction with urban infrastructure [34–38]. In this study, data entries from EVCS locations without complete data were filtered out. For example, if a user review is from an EVCS with no available land use data, we remove that user review. As a result, we retained 2,029 user reviews from 244 unique EVCS locations. For each EVCS location, this study built a 400m buffer around it and quantified each proxy since pedestrians can walk 4.8 km/h (3 mph), 400m (0.25 mi) is the distance that one can travel in 5 minutes [39].

This study utilized parcel-level land use inventory data from the City of Austin Open Data Portal. Each parcel was categorized in accordance with the city's general land use code, as S1 Fig. For each EVCS, this study calculated the land cover area of each land use category. We then calculated the ratio of each land use category's area relative to the total area of the corresponding EVCS's 400m buffer.

Points Of Interest (POIs) are specific destinations or locations that can be uniquely identified and potentially are of interest to EV drives while their cars are plugged in. POI is an emerging source of geospatial information that contains rich, detailed attributes of places, including corner shops, retail stores, public water fountains, parks, and parking lots. POIs have been utilized in many recent studies in urban data science [40–42]. In this paper, we gathered POI data on OpenStreetMap (OSM) via Overpass Turbo. We obtained all available POIs tagged with top-level keys (e.g., amenity, shop, and tourism). Locations of POIs by category are as S2 Fig. For each EVCS, we counted the number of each POI category within its 400m buffer.

We measured economic self-sufficiency by the ratio of residents living below 200% poverty level. For example, if a person's annual income is $40,000, while the poverty level threshold is $25,000, this person counts as living below 200% of poverty level. We utilized census-tract level Ratio of Income to Poverty Level data from the 2024 American Community Survey 5-Year Estimates.

To extract street design elements around each EVCS, we processed each GSV image and calculated the area that each of the following street design elements occupies: Vegetation, Sky, Road, Building, Signage, Person, Vehicle, Sidewalk, Fence, and Bench. This value is known as View Index (VI) and had been used to calculate street design quality indicators such as complexity and walkability [43,44]. Complexity and walkability scores are calculated with Equations 1 and 2.

$$Complexity = \frac{VI_{vegetation} + VI_{person} + VI_{signage} + VI_{vehicle}}{VI_{building} + VI_{road}}$$

(1)

$$Walkability = \frac{VI_{sidewalk} + VI_{fence}}{VI_{road}}$$

(2)

All data collection, processing, and analysis procedures conducted in this study strictly complied with the terms and conditions of the respective data sources. Specifically, the web scraping of publicly accessible EVCS information and user reviews from PlugShare, the extraction of imagery via the Google Maps API, and the utilization of open datasets from the City of Austin Open Data Portal, OpenStreetMap (OSM), and the American Community Survey (ACS) were performed in accordance with their established API usage limits, data sharing policies, and attribution guidelines.

## Methods and AI agent tools

**Sentiment and review categorizing with Generative AI.** A prior study benchmarked multiple generative AI models including Claude, LLaMA, and ChatGPT-4o against conventional NLP-based sentiment analysis [8]. The results showed that generative AI could achieve over 90% accuracy in reproducing traditional sentiment classifications, with ChatGPT-4o consistently outperforming the other models in contextual understanding and classification reliability. These findings validate the applicability of GenAI for analyzing user-generated text in urban infrastructure research and justify the adoption of ChatGPT-4o for this study.

Building on this evidence, this present study employs ChatGPT-4o to analyze and categorize PlugShare user reviews of EVCSs. Each review was processed through structured prompts, enabling the model to assign one or more thematic categories. To ensure methodological transparency and reproducibility, the exact structured prompts designed for sentiment classification and category assignment are provided in full in S1 Table. The categories capture critical aspects of charging experience, including station operation, charging capacity, accessibility and urban environment, parking availability, cost and pricing, technology and network reliability, and miscellaneous feedback (Table 1).

**Spatial pattern analysis.** Before integrating spatial indicators with machine learning models, this study employed Kernel Density Estimation (KDE) to visualize the spatial distribution of EVCSs and identify geographic clusters (hotspots) of specific user experiences. KDE is a widely utilized non-parametric spatial analysis technique that transforms discrete point data, such as individual EVCS locations or user reviews, into a continuous density surface [45].

**Table 1. Category setting for user review analysis.**

| Category | Description |
|---|---|
| Charging Station Operation | Station functionality, service hours, and maintenance conditions |
| Charging Capacity and Performance | Charging speed, congestion levels, and charger efficiency |
| Accessibility and Urban Environment | Location convenience, ease of access, and proximity to urban amenities |
| Parking Availability | Availability of EV-dedicated spaces, parking convenience, and ICE-ing issues |
| Cost and Pricing | Charging fees, membership perks, and pricing transparency |
| Technology and Network | App usability, network reliability, and payment system performance |
| Others | Miscellaneous feedback not covered by other categories |

The density value at any specific spatial location is calculated using the following kernel function:

$$f(x, y) = \frac{1}{nh^2} \sum_{i=1}^{n} K(\frac{d_i}{h})$$

(3)

where $f(x, y)$ represents the estimated density at a given location $(x, y)$, $n$ is the total number of observation points (e.g., reviews in a specific category), $h$ is the bandwidth (or search radius), $K$ is the spatial weight function (kernel), and $d_i$ is the distance between the location $(x, y)$ and the $i$-th observation point [46].

Instead of mapping raw, overlapping data points which can be visually overwhelming in dense urban areas, KDE smooths the spatial data. The bandwidth determines the level of smoothing; a well-calibrated bandwidth filters out localized noise while preserving meaningful macro-level spatial trends [47]. By applying this formula, this study generated continuous heat maps for each sentiment category. This approach allows readers without specialized geospatial knowledge to intuitively identify where specific issues, such as operational failures in peripheral areas or parking shortages in the downtown core, are clustered across Travis County, thereby translating abstract geographic coordinates into actionable urban planning insights.

**GenAI-based image processing.** To process GSV images and obtain VI values, complexity scores, and walkability scores for each street design element, this study implemented an automated procedure assisted by Google's generative AI, Gemini [48]. This study used Gemini 2.5 Flash, the newest multimodal model capable of advanced image understanding including generating semantic segmentation masks. Semantic segmentation is a classic task in the field of computer vision which analyzes an image's context, and label objects present in the image at a pixel level. This technique has been used in recent research including transportation and urban analytics [49,50]. In practice, such tasks are often performed by either building or fine-tuning a neural network. However, traditional deep learning approaches are highly constrained by their fixed taxonomies. They can only recognize a predefined set of classes and require extensive manual annotation and computationally expensive fine-tuning to identify novel or context-specific urban elements.

To overcome this limitation, this study utilized Gemini 2.5 Flash. Recent foundation Multimodal Large Language Models (MLLMs) like Gemini possess zero-shot, open-vocabulary spatial understanding capabilities [51–53]. This allows the model to generate semantic segmentation masks for any street design element described via natural language prompts without requiring specialized task-specific training. To ensure the reliability of this generative approach, we conducted a manual visual inspection on a random subset of 50 GSV images, confirming that the generated masks for vegetation, sky, sidewalks, and built elements consistently aligned with human visual perception. [54]. For each GSV image, this study prompted Gemini to understand the image, create semantic segmentation masks for each street design element, and count the number of pixels in each mask (VI value).

## Random forest

To examine the relationship between various urban features and user experience at EV charging stations (EVCS), this study employed Random Forest (RF) regression models. Seven separate models were constructed, each using a different dependent variable representing aspects of EVCS user experience or spatial context.

Traditional linear modeling approaches, such as Ordinary Least Squares (OLS) regression, assume linear relationships and are highly sensitive to multicollinearity. However, urban environmental variables are often highly correlated and interact in complex, nonlinear ways. We selected RF because it is a robust ensemble learning method that effectively handles non-linear relationships, is resilient to multicollinearity, and does not require feature scaling [55,56]. Most importantly, RF natively computes feature importance, which aligns perfectly with our objective to evaluate the relative influence of multiple predictors rather than solely maximizing predictive accuracy.

Table 2 summarizes the structure of the six RF models, including their dependent and independent variables. The RF models were trained using scikit-learn with default hyperparameters. For each model, the top 15 predictors were extracted and compared across dependent variables to identify patterns in variable influence. Detailed basic statistics for all input variables are provided in Section 4.2 and Supporting information (S2 and S3 Tables).

# Results

## Spatial patterns of EV charging stations and user reviews

This section presents a descriptive analysis of EVCSs and user reviews in Travis County, using kernel density and category-based visualizations. Fig 2 shows the kernel density of 363 EVCSs, with the densest clusters located in downtown Austin. The Domain area, a commercial hub in North Austin, displays the second-highest density. Most EVCSs are concentrated along Austin's central north–south corridor, with additional extensions westward toward the lake. This distribution reflects higher density in commercial and residential cores, while peripheral and rural areas remain sparsely covered. Also, Fig 2 illustrates the number of reviews by category in a lollipop plot. Charging Station Operation (730 reviews) is the most frequently discussed topic, followed by Capacity and Performance (398 reviews), and Technology and Network (302 reviews). Accessibility and Urban Environment (109 reviews) and Parking Availability (271 reviews) receive fewer mentions, suggesting that user concerns are primarily directed toward technical reliability rather than contextual integration.

Fig 3 provides kernel density maps of user reviews by category. Operation issues are found both in downtown and suburban areas, indicating that reliability challenges extend beyond the urban core. Capacity and Performance and Technology and Network follow similar dispersed patterns, reflecting congestion and connection concerns that are county-wide rather than localized.

**Table 2. Summary of random forest regression models.**

| Model Structure | Variables Used in the Analysis |
|---|---|
| Dependent Variables (DV) (A separate model was built for each of the 6 categories) | 1. Charging Station Operation<br>2. Charging Capacity and Performance<br>3. Accessibility and Urban Environment<br>4. Parking Availability<br>5. Cost and Pricing<br>6. Technology and Network |
| Independent Variables (IVs) (Identical set applied across all 6 models) | • Land Use Composition: Ratios of Commercial, Residential, Civic, Industrial, Open Space, etc.<br>• Points of Interest (POIs): Counts of Amenities, Shops, Tourism, etc.<br>• Socioeconomic Context: Ratio of residents living below 200% of the poverty level.<br>• Google Street View (GSV) Elements: View Index (VI) of Sidewalk, Vegetation, Sky, Building, Road, etc.<br>• Derived Street Quality Scores: Walkability Score, Complexity Score. |

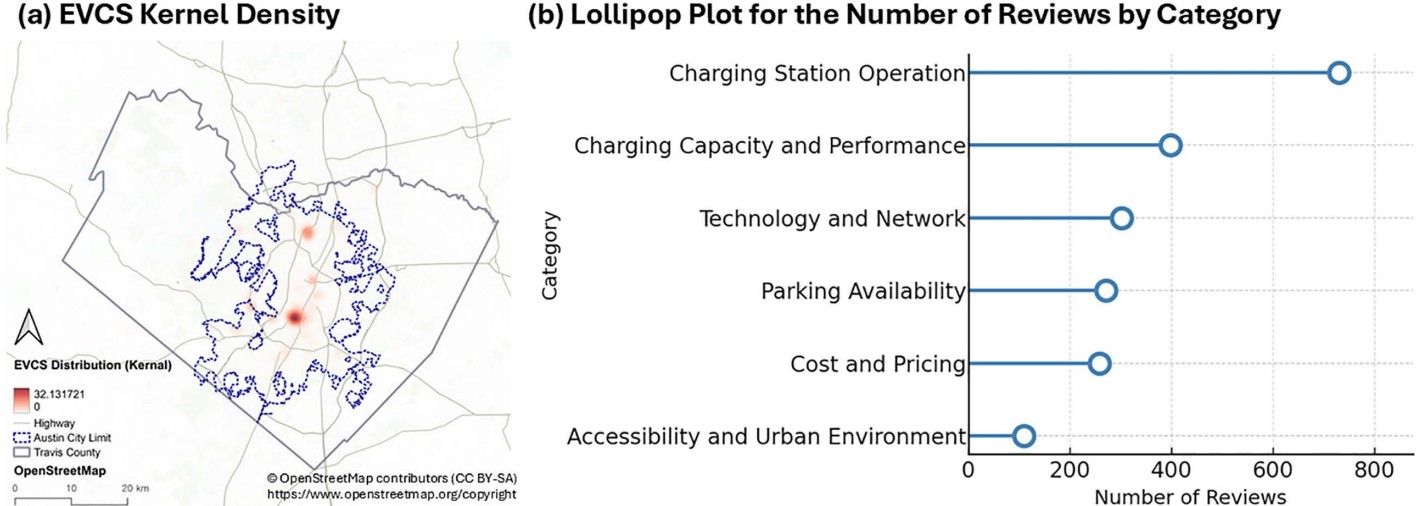

**Fig 2. EV Charging Stations and User Review Status: (a)** EVCS Kernel Density, **(b)** Lollipop Plot for the Number of Reviews by Category. The base map was created using OpenStreetMap data (https://www.openstreetmap.org/copyright) under a CC BY 4.0 license.

In contrast, reviews related to Accessibility and Urban Environment and Parking Availability concentrate in downtown and dense commercial zones. These clusters suggest that users in high-density environments evaluate EVCSs not only for charging functionality but also for their integration with walkability, nearby amenities, and parking design. Cost and Pricing reviews, meanwhile, tend to cluster around shopping centers and dining areas, highlighting the linkage between EV charging and consumer contexts.

These findings demonstrate a geographic differentiation of user concerns. Technical and operational challenges appear spatially widespread, underscoring the need for improved infrastructure reliability across suburban and peripheral zones. In contrast, access- and context-related issues are concentrated in urban centers, pointing to planning needs that integrate EVCSs more seamlessly with mobility, land use, and economic activities.

### GenAI-based urban environment analysis

An example of the input and output of this procedure is shown in S3 Fig. Each GSV image was spatially joined with EVCs locations; if multiple GSV images were present within a 500m radius from an EVCS, we calculated the average VI value, complexity, and walkability. Detailed parcel-level land use categories and summary of input data is in Appendices (S2 and S3 Tables).

Figs 4 and 5 visualize the spatial context of EVCS locations using violin plots. Fig 4 summarizes parcel-level land-use composition within the EVCS catchments, and Fig 5 depicts the distribution of visually derived street-scene features from GSV imagery. As described earlier, each GSV image was spatially joined to EVCS points; when multiple images fell within 500 m of a station, we computed catchment-level means for the semantic-segmentation outputs (and for VI, complexity, and walkability as reported elsewhere).

The land-use distribution reveals that EVCSs are most embedded in commercial and residential settings. Single-family categories (including large-lot single family) exhibit higher central mass with wide dispersion, indicating that many stations are placed in predominantly residential catchments, while a nontrivial subset sits in areas with very high residential dominance. Multi-family shares, by contrast, tend to be lower on average but show long upper tails, suggesting localized clusters of EVCSs in denser housing contexts. Commercial and office uses present low medians, consistent with stations

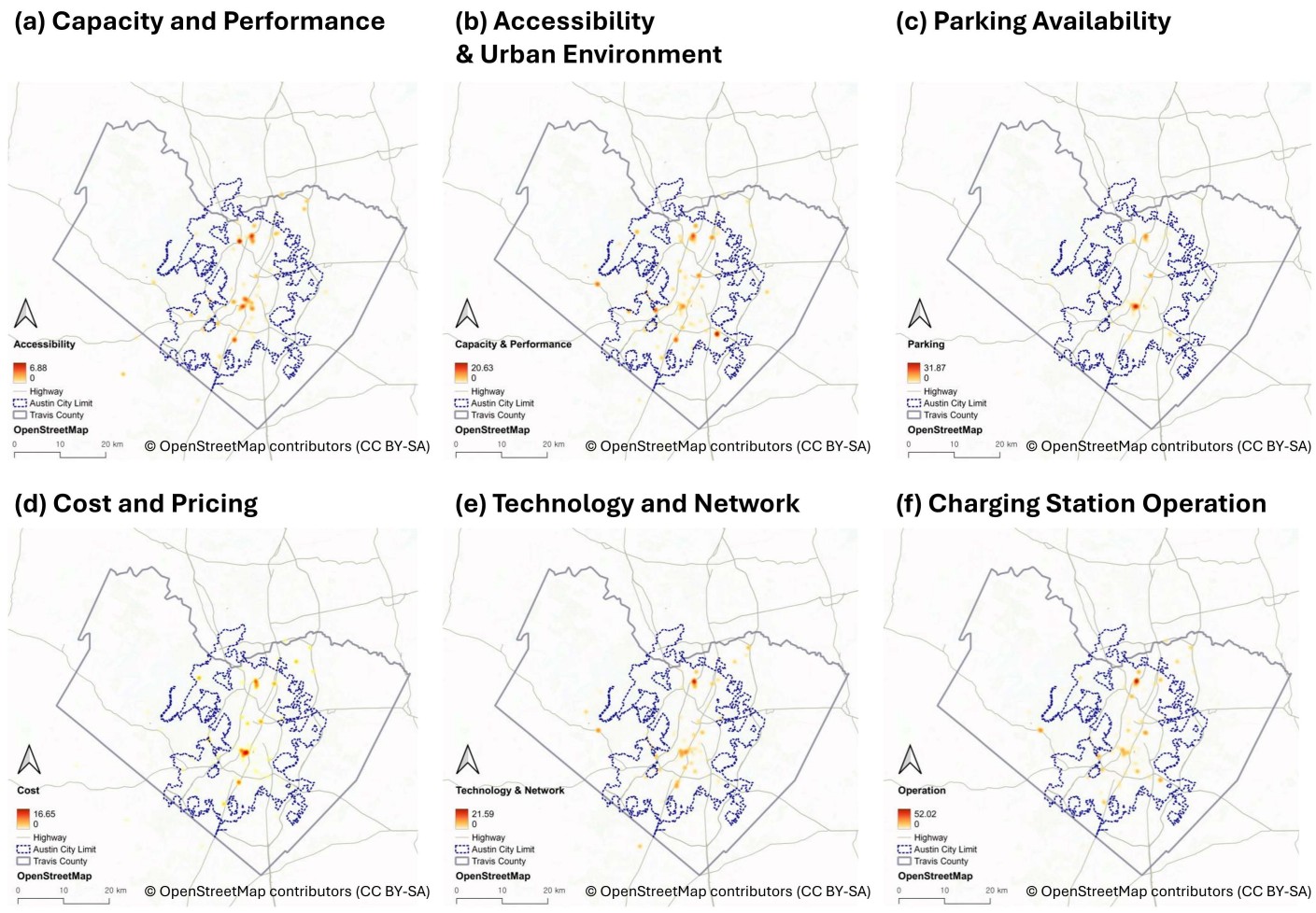

**Fig 3. Kernel Density of User Reviews by Category: (a)** Charging Capacity and Performance, **(b)** Accessibility and Urban Environment, **(c)** Parking Availability, **(d)** Cost and Pricing, **(e)** Technology and Network, **(f)** Charging Station Operation. The base map was created using OpenStreetMap data (https://www.openstreetmap.org/copyright) under a CC BY 4.0 license.

that are generally outside purely commercial cores but are present in selected activity centers and employment nodes. Mixed-use also appears with a low central tendency yet a thick upper tail, implying that a minority of stations serve genuinely mixed urban fabrics. Open space and parks show modest but heterogeneous presence; some EVCSs about green amenities, whereas many do not. Transportation, utilities, right-of-way, and industrial classes generally concentrate near zero for most stations and flare only in specific cases, indicating that EVCS siting is not primarily co-located with large infrastructure or industrial parcels. Water and resource-extraction categories are effectively negligible for most sites. The land-use pattern characterizes EVCSs as everyday neighborhood infrastructure: predominantly residential in catchment composition, with selective integration into commercial/mixed nodes and occasional adjacency to parks.

The GSV-based environmental features complement this picture by describing how the street space itself is composed around EVCSs. Road and vehicle masks show strong central mass, indicating that stations are typically situated along well-trafficked corridors. Building shares are consistently present at moderate to high levels, reflecting a built-up context rather than peripheral or exurban sites. Sidewalk elements appear across many catchments, suggesting that pedestrian access is commonly available even where carriageway space dominates. By contrast, sky and vegetation masks

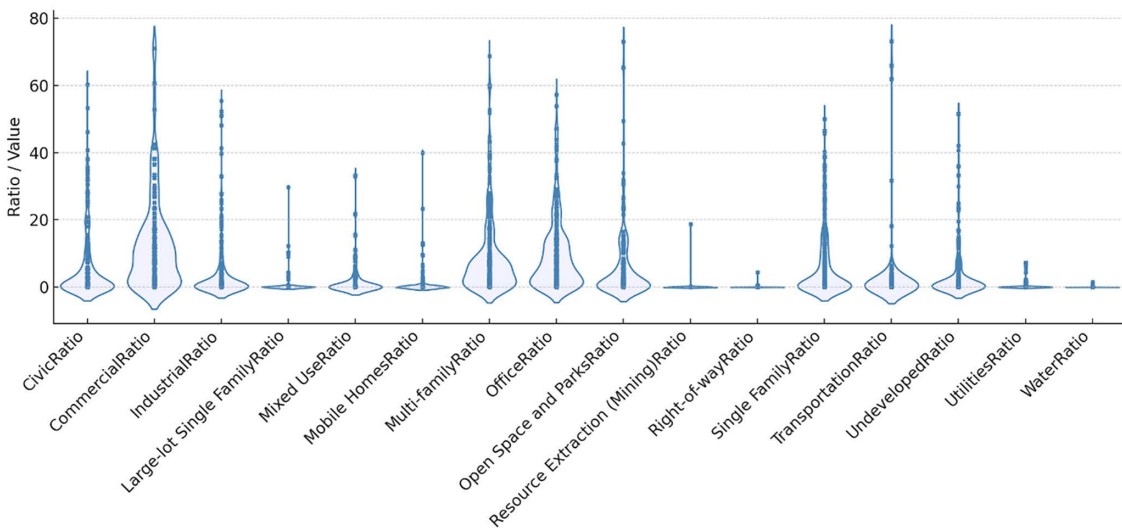

**Fig 4. Violin Plot of Land Use Variables.**

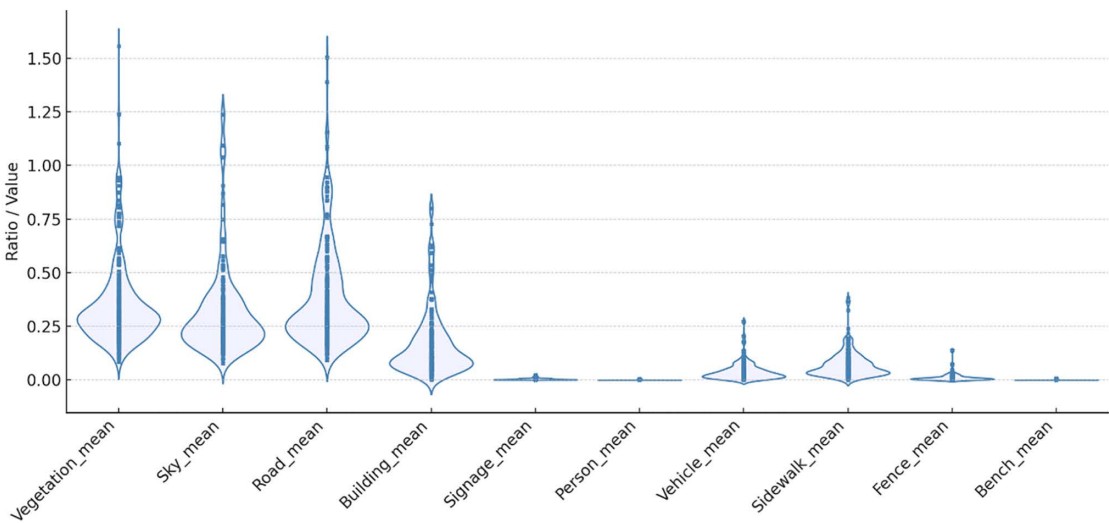

**Fig 5. Violin Plot of Google Street View Panorama Variables.**

exhibit high variability: some stations sit in tight, visually enclosed streetscapes with little sky or greenery visible, while others occupy more open or greener edges, hinting at diverse local morphologies. Low but occasional spikes in signage, benches, and fences point to a subset of locations with richer streetscape furniture or controlled frontages. Person masks are sparse, as expected from the image source and timing, and do not systematically characterize station areas. The GSV distributions portray EVCSs as closely coupled to vehicular thoroughfares and building fronts, with substantial heterogeneity in openness and greenery and with generally present pedestrian infrastructure.

## User experience and the relationship with surrounding urban environments

To identify which characteristics of the built environment most influence EVCS user experience, this study implemented Random Forest (RF) regression models, each corresponding to a distinct dimension of user perception, including parking availability, accessibility, station operation, technology and network, cost and pricing, and charging capacity. For each model, this study extracted the top 15 most influential predictors. Detailed feature importance plots for each category are presented in S4 Fig, while Fig 6 integrates these results into a comparative rank heatmap for cross-model analysis.

The heatmap reveals two clear patterns. First, several predictors consistently appear in the top ranks across multiple models. Variables derived from Google Street View (GSV) imagery such as Sidewalk, Vegetation, and Sky emerge as strong common predictors, alongside contextual variables like Civic land use ratio and poverty ratio. These features capture openness, greenery, walkability, and socio-spatial context, underscoring that user experience at EVCSs is shaped not only by technical infrastructure but also by qualities of the surrounding built environment.

Second, the heatmap highlights important divergences across perception dimensions. Parking Availability and Accessibility are strongly associated with Sidewalk, Vegetation, Walkability, and Open Space, suggesting that users interpret ease of parking and accessibility through visual spaciousness and pedestrian-friendly settings. In contrast, Technology and Network and Charging Capacity emphasize Complexity score, building presence, and dense land-use ratios, pointing to a closer link between technical reliability perceptions and more built-up or visually complex contexts. Cost and Pricing, by comparison, draws on a wider mix of spatial and socioeconomic variables, reflecting their multifaceted nature.

The RF results indicate that certain environmental attributes, such as walkability, greenery, and openness, are universally influential across multiple user perceptions, while other predictors are dimension-specific, reflecting the distinct ways users evaluate accessibility versus technical performance. Integrating EVCS into legible, walkable, and visually appealing environments may therefore enhance user satisfaction broadly, while improving perceptions of technical reliability requires attention to infrastructural and built-form contexts.

## Discussion and conclusion

The growing adoption of EVCSs has elevated them from technical nodes of transportation electrification to socially and spatially embedded infrastructure within urban mobility networks. As EV usage expands, user experiences at charging stations are becoming a pivotal determinant of accessibility, satisfaction, and broader adoption. Understanding how users perceive EVCSs, and which environmental and contextual factors shape these perceptions, is essential for ensuring both technical performance and integration into everyday urban life.

This study addressed this need by combining large-scale user review analysis with a generative AI–assisted spatial framework. By applying GenAI to more than 4,000 user-generated reviews and linking them with parcel-level land use, points of interest, socioeconomic indicators, and GSV imagery, the study provides new insight into how urban environments condition user experiences. The RF models revealed that spatial and visual features, including sidewalks, vegetation, sky openness, and surrounding amenities, consistently rank among the most influential predictors of user satisfaction. These results suggest that EVCS usability is not determined by operational factors alone, but also by perceptual qualities of the surrounding urban fabric.

The results highlight both commonalities and divergences across user perception categories. Attributes such as sidewalk presence, walkability, and openness emerge as universally influential, shaping satisfaction with both parking and accessibility. In the context of urban design, the proportion of sky in street-level imagery serves as a critical proxy for visual openness and the sense of enclosure. A higher sky ratio often indicates environments that are less visually oppressive, which can enhance perceived safety and psychological comfort for pedestrians and users navigating the station. This indicates that users interpret availability and convenience not only through functional attributes, but also through environmental cues that signal legibility, comfort, and human-scale design. On the other hand, technology and charging capacity places greater

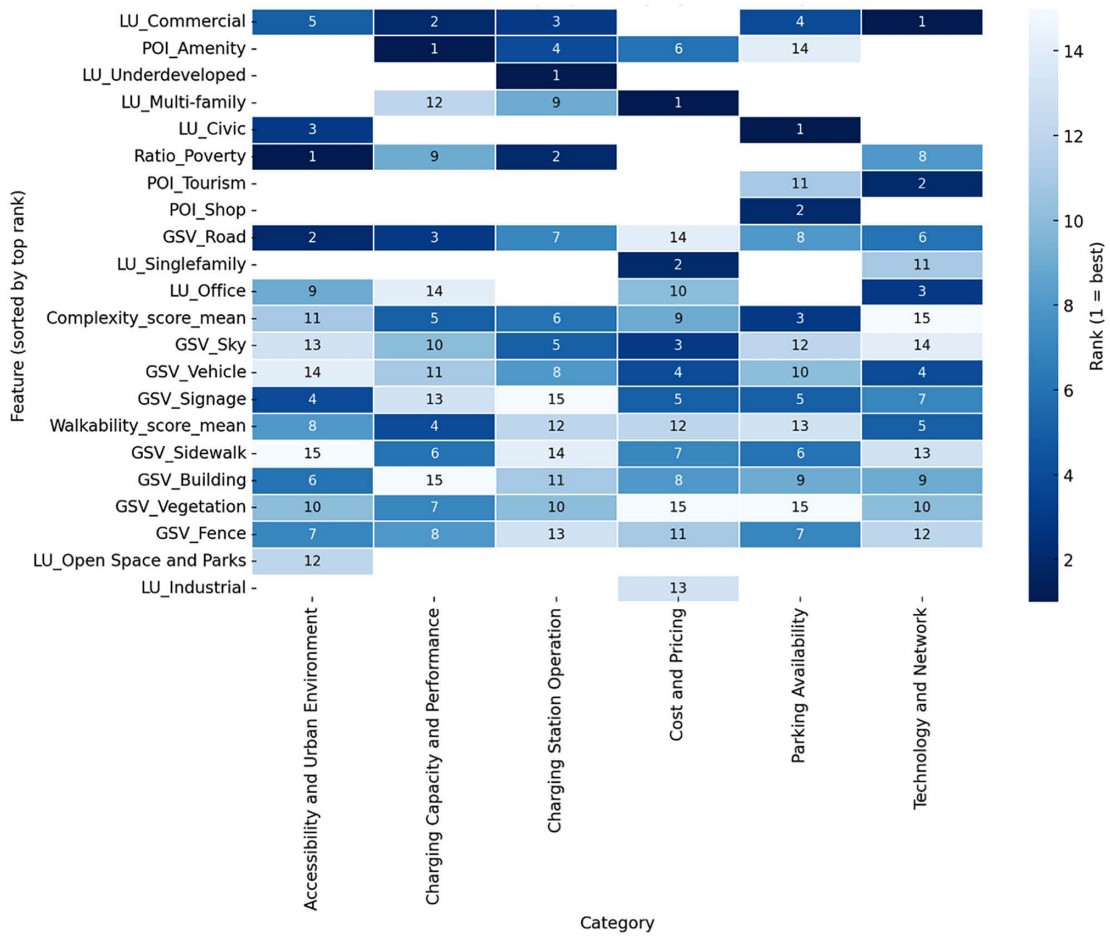

**Fig 6. Feature Importance Rank Heatmap by Category.**

emphasis on built density, building presence, and environmental complexity, suggesting that perceptions of reliability and technical performance are more closely tied to infrastructural and visually complex contexts. Cost and pricing perceptions exhibit a more diffuse set of predictors, reflecting their dependence on both environmental and socioeconomic conditions.

These findings carry important implications for EVCS planning and urban design. First, the results underline the necessity of embedding EVCSs into walkable, green, and visually open environments to enhance user satisfaction. Crucially, the high predictive importance of variables such as sidewalks, walkability, and commercial land use reflects the unique behavioral patterns of EV users. Unlike traditional gas stations where users remain with their vehicles, EV charging requires longer dwelling times, prompting many users to leave the station to shop, eat, work, or rest. Consequently, the immediate built environment is evaluated not merely as a waiting area, but as a pedestrian gateway to surrounding urban functions. This explains why macro-level connectivity and proximity to amenities so strongly dictate user satisfaction. Second, the analysis demonstrates that user experience is context-sensitive: suburban or peripheral locations may require greater focus on operational reliability, while dense urban centers demand attention to parking design and pedestrian access. Recognizing this spatial heterogeneity is essential for tailoring EVCS strategies to different urban forms.

While this study advances a novel framework linking user reviews, generative AI–derived spatial indicators, and machine learning models, several limitations should be acknowledged. First, the analysis is geographically limited to

Travis County, Texas, which may restrict the generalizability of the findings to regions with different urban forms, cultural expectations, or EV adoption rates. Second, the spatial features extracted from GSV and parcel-level data capture the built environment conditions at a single point in time; they do not account for temporal variation in factors such as congestion, vegetation, or amenity turnover, which may influence user perceptions in dynamic ways. Third, the street-level visual features, such as the sky variable, capture the macro-level streetscape but may not fully account for micro-scale site designs, such as weather-protecting canopies or awnings, which could also affect sky visibility and user comfort. Furthermore, the relevance of the immediate built environment may vary depending on user behavior. Unlike refueling at a gas station, EV charging takes longer, meaning many users leave their vehicles to visit nearby amenities rather than staying at the station. For these users, the broader neighborhood context may matter more than the immediate visual environment of the charger itself. Future studies should explicitly distinguish between "stay" and "leave" charging behaviors to better understand how contextual needs shift based on user activity.

Building on this foundation, future research should extend the framework in the following directions. The first is to evaluate the economic impacts of EVCS siting and user experience. By linking environmental predictors and user satisfaction with commercial activity indicators, such as retail sales, restaurant revenues, or dwell time, it would be possible to assess whether positive charging experiences translate into measurable economic spillovers for nearby businesses. Such evidence could inform urban development strategies that view EVCSs not only as mobility infrastructure but also as catalysts for local economic vitality. The second direction is to examine the operational and financial performance of EVCSs in relation to their spatial context. Understanding whether stations embedded in more walkable, amenity-rich, or visually appealing environments generate higher utilization rates, customer retention, or revenue would provide valuable insights for both public agencies and private operators.

By integrating human experience, urban context, and economic outcomes, future research can move beyond assessing EVCS usability in isolation to address broader questions of sustainability, and economic viability. Such an expanded perspective will be critical as cities worldwide seek to design charging networks that are not only technically reliable but also socially embedded and economically beneficial.

## Supporting information

**S1 Fig. Filtered EVCS locations and parcel-level land use (The base map was created using OpenStreetMap data (https://www.openstreetmap.org/copyright) under a CC BY 4.0 license.**).
(PNG)

**S2 Fig. Location of OSM POIs with top-level keys (amenity, shop, and tourism (The base map was created using OpenStreetMap data (https://www.openstreetmap.org/copyright) under a CC BY 4.0 license.**).
(PNG)

**S3 Fig. An example of semantic segmentation performed with Google Gemini Flash 2.5; (a): Original Street View Image, (b): Street View Image with Segregation Masks.**
(PNG)

**S4 Fig. Feature Importance by User Experience Categories.**
(PNG)

**S1 Table. Prompt Design for Sentiment Analysis.**
(DOCX)

**S2 Table. Parcel-level Land Use Categories.**
(DOCX)

**S3 Table. Summary of Input Data.**
(DOCX)

## Author contributions

**Conceptualization:** Ahyoung Chang.

**Data curation:** Ahyoung Chang.

**Formal analysis:** Ahyoung Chang.

**Funding acquisition:** Junfeng Jiao.

**Investigation:** Ahyoung Chang.

**Methodology:** Ahyoung Chang, Seung Gyu Baik.

**Resources:** Junfeng Jiao.

**Software:** Junfeng Jiao.

**Supervision:** Junfeng Jiao.

**Validation:** Ahyoung Chang.

**Visualization:** Ahyoung Chang, Seung Gyu Baik.

**Writing – original draft:** Ahyoung Chang, Seung Gyu Baik.

**Writing – review & editing:** Ahyoung Chang, Junfeng Jiao.

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
