## [Decision Letter · Decision Letter 0]

3 Mar 2026

PONE-D-26-00206How Urban Environment Shapes EV Charging Experience in Travis County, TexasPLOS One

Dear Dr. Chang,

Thank you for submitting your manuscript to PLOS ONE. After careful consideration, we feel that it has merit but does not fully meet PLOS ONE’s publication criteria as it currently stands. Therefore, we invite you to submit a revised version of the manuscript that addresses the points raised during the review process.

We look forward to receiving your revised manuscript.

Kind regards,

Junghwan Kim

Academic Editor

PLOS One

Journal Requirements:

2. "In your Methods section, please include additional information about your dataset and ensure that you have included a statement specifying whether the collection and analysis method complied with the terms and conditions for the source of the data.

3. Please note that PLOS One has specific guidelines on code sharing for submissions in which author-generated code underpins the findings in the manuscript. In these cases, we expect all author-generated code to be made available without restrictions upon publication of the work. Please review our guidelines at https://journals.plos.org/plosone/s/materials-and-software-sharing#loc-sharing-code and ensure that your code is shared in a way that follows best practice and facilitates reproducibility and reuse.

“This research was supported by the NSF Grants (2125858, 2236305), The MITRE  Corporation and UT Good Systems.”

6. We note that Figures 1, 2, S1 & S2 in your submission contain [map/satellite] images which may be copyrighted. All PLOS content is published under the Creative Commons Attribution License (CC BY 4.0), which means that the manuscript, images, and Supporting Information files will be freely available online, and any third party is permitted to access, download, copy, distribute, and use these materials in any way, even commercially, with proper attribution. For these reasons, we cannot publish previously copyrighted maps or satellite images created using proprietary data, such as Google software (Google Maps, Street View, and Earth). For more information, see our copyright guidelines: http://journals.plos.org/plosone/s/licenses-and-copyright.

a. You may seek permission from the original copyright holder of Figure(s) [#] to publish the content specifically under the CC BY 4.0 license.

8. We notice that your supplementary figures are uploaded with the file type 'Figure'. Please amend the file type to 'Supporting Information'. Please ensure that each Supporting Information file has a legend listed in the manuscript after the references list.

Reviewers' comments:

Reviewer's Responses to Questions

**Comments to the Author**

1. Is the manuscript technically sound, and do the data support the conclusions?

Reviewer #1: Partly

Reviewer #2: Yes

2. Has the statistical analysis been performed appropriately and rigorously? 

Reviewer #1: Yes

Reviewer #2: Yes

3. Have the authors made all data underlying the findings in their manuscript fully available?

Reviewer #1: Yes

Reviewer #2: Yes

4. Is the manuscript presented in an intelligible fashion and written in standard English?

Reviewer #1: Yes

Reviewer #2: Yes

5. Review Comments to the Author

Reviewer #1: This research positions itself as a spatial analysis of EVCS user sentiment, which is a novel and potentially valuable approach. Overall, the manuscript meets the general quality standards of PLOS One and addresses a timely topic of strong interest to transportation electrification researchers as well as the broader business community. However, to be suitable for publication, the study needs to more clearly reference prior work and more rigorously justify its methodological choices.

Literature Review

Page 5. The authors state, “Furthermore, most existing studies have emphasized model performance and spatial clustering of sentiment, while offering limited insight into the causal mechanisms linking user experience to urban context.” This is a substantial claim and requires supporting references. It is unclear which specific “existing studies” the authors are referencing.

Page 6. The authors should be cautious when stating, “AI has increasingly been used for demand forecasting and system monitoring.” This claim is not adequately supported by the text. While a few experimental case studies are mentioned, the statement implies widespread practitioner adoption, which has not been established.

Page 7. The statement, “Sentiment analysis of user-generated reviews reveals that perceptions of safety, walkability, and comfort in the charging environment often surface alongside operational feedback,” also requires appropriate citations.

Materials and Methods

Section 3.1.1, Study Area. The rationale for selecting Travis County as the study area, rather than the entire Austin Metropolitan Area, is unclear. While the text references the county’s urban core in Austin, it provides little discussion of the suburban and rural portions of the county. Given the heterogeneity of the region, additional explanation is needed to provide adequate contextual grounding.

Section 3.1.2, Page 9. The authors should explain why land use, points of interest, economic self sufficiency, and street design elements were selected as proxy variables rather than alternative measures. These choices should be explicitly justified to support replicability. References to the existing literature would strengthen this justification.

Section 3.1.2, Page 10. The manuscript should specify which year of American Community Survey data was used and whether the data represent 1 year, 3 year, or 5 year estimates.

Section 3.2.1. The “structured prompts” used for sentiment analysis should be provided, as this information is necessary for replicability.

Section 3.2.2. Gemini appears to be treated as a black box that is assumed to be capable of semantic segmentation without sufficient validation by the authors. No related research is cited demonstrating semantic segmentation using Gemini 2.5 Flash. The authors should explain why a more traditional deep learning approach was not considered for this task.

Section 3.2.3. The authors should include a table or figure summarizing the seven random forest regression models and their associated variables. In addition, the choice of random forest regression should be justified relative to alternative modeling approaches.

Results and Discussion

At a high level, the results provide new and interesting insights into the spatial dimensions of EVCS user experiences that have not been widely explored in prior research. The authors should be commended for this contribution. However, the figures are consistently grainy and difficult to interpret. Higher resolution images with larger labels and added contextual elements, such as network centerlines and municipal boundaries, would greatly improve readability and interpretability.

It is also unclear how “sky” constitutes a meaningful street design element. This variable should likely be removed from the analysis, as the amount of visible sky in Google Street View imagery is unlikely to meaningfully influence user experiences at EVCS locations. A more relevant variable from the user perspective would be whether the charging station is covered or uncovered by an awning or overhang that provides weather protection. In addition, the images appear to be captured near the EVCS sites rather than directly in front of them, further limiting the interpretive value of the sky variable.

Relatedly, an EVCS station does not function in the same way as a bus stop. Many users do not remain at the station while their vehicles charge. Instead, EVCS locations often serve as places where users leave their vehicles while they shop, work, eat, or rest. In this context, the immediate built environment around the station may be of limited relevance to the user experience. This distinction should be explored more explicitly in relation to the study’s findings.

Finally, the authors should revisit their visualization methods and provide a more thorough explanation of their use of kernel density estimation and hotspot analysis in the methodology section. These techniques may not be familiar to readers outside transportation geography and spatial analysis fields, and additional methodological detail would improve accessibility and transparency.

Reviewer #2: This article is well written and presented in a simplified form for the general reader. The authors did a detailed analysis and presented the information in an easy-to-understand manner. The authors applied a methodological approach and applying the emerging A.I technology into their research to convey the message to the public.

6. PLOS authors have the option to publish the peer review history of their article (what does this mean?). If published, this will include your full peer review and any attached files.

Reviewer #1: No

Reviewer #2: No

---

## [Author Response · Author response to Decision Letter 1]

26 Mar 2026

Please find our detailed, point-by-point responses to all editor and reviewer comments in the attached "Response to Reviewers" document. Thank you so much.

---

## [Decision Letter · Decision Letter 1]

3 May 2026

How Urban Environment Shapes EV Charging Experience in Travis County, Texas

PONE-D-26-00206R1

Dear Dr. Chang,

We’re pleased to inform you that your manuscript has been judged scientifically suitable for publication and will be formally accepted for publication once it meets all outstanding technical requirements.

Kind regards,

Junghwan Kim

Academic Editor

PLOS One

Additional Editor Comments (optional):

Reviewers' comments:

Reviewer's Responses to Questions

**Comments to the Author**

1. If the authors have adequately addressed your comments raised in a previous round of review and you feel that this manuscript is now acceptable for publication, you may indicate that here to bypass the “Comments to the Author” section, enter your conflict of interest statement in the “Confidential to Editor” section, and submit your "Accept" recommendation.

Reviewer #1: All comments have been addressed

Reviewer #2: All comments have been addressed

2. Is the manuscript technically sound, and do the data support the conclusions?

Reviewer #1: Yes

Reviewer #2: Yes

3. Has the statistical analysis been performed appropriately and rigorously? 

Reviewer #1: Yes

Reviewer #2: Yes

4. Have the authors made all data underlying the findings in their manuscript fully available?

Reviewer #1: Yes

Reviewer #2: Yes

5. Is the manuscript presented in an intelligible fashion and written in standard English?

Reviewer #1: Yes

Reviewer #2: Yes

6. Review Comments to the Author

Reviewer #1: The authors have done a good job of addressing the reviewer comments. Though some of the figures are still a bit blurry, this can be rectified in the final proofing process. I believe that this work meets the requirements of a publishable PLOS One manuscript as it is now written.

Reviewer #2: The authors addressed the questions highlighted and improve the overall flow of the paper addressing most of the questions previously not clear.

7. PLOS authors have the option to publish the peer review history of their article (what does this mean?). If published, this will include your full peer review and any attached files.

Reviewer #1: No

Reviewer #2: No

---

## [Editor Report · Acceptance letter]

PONE-D-26-00206R1

PLOS One

Dear Dr. Chang,

I'm pleased to inform you that your manuscript has been deemed suitable for publication in PLOS One. Congratulations! Your manuscript is now being handed over to our production team.

Kind regards,

on behalf of

Dr. Junghwan Kim

Academic Editor

PLOS One